# Lys29-linkage of ASK1 by Skp1−Cullin 1−Fbxo21 ubiquitin ligase complex is required for antiviral innate response

Zhou Yu[1,2,3†], Taoyong Chen[2*†], Xuelian Li[2†], Mingjin Yang[2,3], Songqing Tang[1], Xuhui Zhu[2], Yan Gu[2], Xiaoping Su[2], Meng Xia[3], Weihua Li[4], Xuemin Zhang[4], Qingqing Wang[1], Xuetao Cao[2,3], Jianli Wang[1*]

[1]Institute of Immunology, Zhejiang University School of Medicine, Hangzhou, China; [2]National Key Laboratory of Medical Immunology and Institute of Immunology, Second Military Medical University, Shanghai, China; [3]National Key Laboratory of Medical Molecular Biology and Department of Immunology, Institute of Basic Medical Sciences, Chinese Academy of Medical Sciences, Beijing, China; [4]Institute of Basic Medical Sciences, National Center of Biomedical Analysis, Beijing, China

**Abstract** Protein ubiquitination regulated by ubiquitin ligases plays important roles in innate immunity. However, key regulators of ubiquitination during innate response and roles of new types of ubiquitination (apart from Lys48- and Lys63-linkage) in control of innate signaling have not been clearly understood. Here we report that F-box only protein Fbxo21, a functionally unknown component of SCF (Skp1–Cul1–F-box protein) complex, facilitates Lys29-linkage and activation of ASK1 (apoptosis signal-regulating kinase 1), and promotes type I interferon production upon viral infection. Fbxo21 deficiency in mice cells impairs virus-induced Lys29-linkage and activation of ASK1, attenuates c-Jun N-terminal kinase (JNK) and p38 signaling pathway, and decreases the production of proinflammatory cytokines and type I interferon, resulting in reduced antiviral innate response and enhanced virus replication. Therefore Fbxo21 is required for ASK1 activation via Lys29-linkage of ASK1 during antiviral innate response, providing mechanistic insights into non-proteolytic roles of SCF complex in innate immune response.

*For correspondence: chenty@ immunol.org (TC); jlwang@zju.edu. cn (JW)

†These authors contributed equally to this work

## Introduction

Innate immune system serves as the first line defense of invading microorganisms and endogenous danger signals. Host cells can recognize pathogen-associated molecular patterns (PAMPs) of invading pathogens via pattern recognition receptors (PRRs), initiate signaling pathways to regulate the expression of proinflammatory cytokines and type I interferons (IFN α/β) (*Yin et al., 2015*). Upon virus infection, host cells can also recognize viral components, especially nucleic acids derived from viral genome or produced during viral life cycle (*Gürtler and Bowie, 2013*). Viral RNAs are mainly detected by RIG-I-like receptors (such as RIG-I and MDA5) that induce type I IFNs via MAVS (mitochondrial antiviral-signaling protein, also known as IPS1, Cardif and VISA)–dependent activation of TANK-binding kinase 1 (TBK1)–interferon regulatory factor 3 (IRF3) signaling pathway (*Chan and Gack, 2015*; *Yoneyama et al., 2015*). On the other hand, viral DNAs are detected by multiple DNA sensors that induce type I IFNs mainly via stimulator of interferon genes (STING, also known as MITA, MPYS and ERIS)–dependent TBK1–IRF3 signaling pathway (*Paludan and Bowie, 2013*; *Cai et al., 2014*). Despite of these findings, the innate response to viruses has not been completely understood. Studies are still required to elucidate detailed signaling and regulatory mechanisms initiated by PRRs in antiviral innate response.

**eLife digest** The innate immune system is the body's first line of defense against being infected by viruses and other microbes. Upon recognizing a virus, host cells trigger the innate immune response in an effort to eliminate the threat. However, innate immune responses must be carefully controlled because an excessive response can cause inflammation that harms the body.

The innate immune response involves a variety of cells and processes that are each activated through a series of communication systems called signaling pathways. While much has been learned about which parts of a virus trigger the innate immune response, it is not clear how the immune response to the virus is controlled.

It has been suggested that a process known as ubiquitination could be involved in regulating the activity of signaling pathways that activate the innate immune response. During ubiquitination, enzymes attach a small molecule called ubiquitin to a specific target protein. Ubiquitin often acts as a label that targets a particular protein for destruction. Enzymes called E3 ubiquitin ligases play central roles in identifying specific target proteins for ubiquitination. Some of these enzymes consist of a single protein unit that acts alone, but other E3 ubiquitin ligases are formed by groups (or "complexes") of several proteins working together.

Members of the F-box only protein family are components of some ubiquitin ligase complexes. Here, Yu et al. used a "microarray" technique to assess which F-box only proteins in mice are produced during an immune response to two viruses. The experiments identified an F-box protein called Fbxo21 as a potential candidate for a role in regulating the innate immune response.

Additional experiments revealed that Fbxo21 is involved in adding ubiquitin to a specific location on a signaling protein called ASK1, which is known to be crucial for innate immune responses. Instead of targeting ASK1 for destruction, this ubiquitination activates ASK1. Therefore, Yu et al.'s findings demonstrate that Fbxo21 plays an important role in regulating innate immune responses. A future challenge is to investigate exactly how ASK1 is activated by the ubiquitin.

Ubiquitin (Ub) modification is one type of the crucial posttranslational modification mechanisms, and has been implicated in regulation of both innate and adaptive immunity (*Jiang and Chen, 2011*; *Zinngrebe et al., 2014*). As compared to the typical single subunit E3 ligases (such as RING-finger type and HECT type E3 ligases), the Cullin-RING multisubunit ligase complexes are assembled on the cullin backbones and have been implicated in diverse processes (*Skaar et al., 2013*; *2014*). S phase kinase-associated protein 1 (Skp1)–cullin 1 (Cul1)–F-box protein (SCF) complex is example of the Cullin-RING multisubunit ligase complexes (*Skaar et al., 2013*; *2014*). In this complex, Skp1 binds to F-box domain of many F-box proteins. More than sixty F-box proteins have been identified up to date, and they have been classified into three categories: those with WD40 domain (Fbxw), those with leucine-rich repeats (Fbxl) and those with other diverse domains (Fbxo) (*Jin et al., 2004*; *Skaar et al., 2013*; *2014*). Despite great achievements have been made for elucidating the roles of SCF complexes in cancer (*Skaar et al., 2014*), the function of SCF in immunity remains poorly understood. Possibly the best characterized SCF complexes associated with immunity are the F-box– and WD40 repeat–containing Fbxw1 (also known as β-Trcp1, Fbw1a and Fwd1) and Fbxw11 (also known as β-TrCP2, Fbw11, Fbxw1b, Fbx1b and Hos) complexes that have been implicated in regulating NF-κB signaling by degrading IκBα or processing of p100 (also known as NFκB2) and p105 (also known as NFκB1) (*Yaron et al., 1998*; *Shirane et al., 1999*; *Spencer et al., 1999*; *Tan et al., 1999*; *Wu and Ghosh, 1999*; *Orian et al., 2000*; *Fong and Sun, 2002*; *Lang et al., 2003*; *Amir et al., 2004*). Since Fbxo proteins usually contain unidentified variable domains in addition to the F-box domain, the functions of Fbxo proteins remain largely unknown, especially in regulation of immunity.

Apoptosis signal-regulating kinase 1 (ASK1, also known as Map3k5) is a mitogen-activated protein kinase (MAPK) kinase kinase that plays pivotal roles in stress and immune responses (*Ichijo et al., 1997*; *Shiizaki et al., 2013*). ASK1 can activate MKK4/6 and MKK3/7, leading to activation of c-Jun N-terminal kinases (JNK1/2) and p38 MAPKs (*Ichijo et al., 1997*; *Shiizaki et al., 2013*). ASK1 is crucial for apoptosis in response to multiple stresses and factors (*Ichijo et al., 1997*; *Sagasti et al., 2001*; *Tobiume et al., 2001*; *Nishitoh et al., 2002*; *Maruoka et al., 2003*;

*Takeda et al., 2004*; *Matsuzawa et al., 2005*; *Shiizaki et al., 2013*). ASK1 has also been implicated in Toll-like receptor 4 (TLR4)-triggered innate responses (*Matsuzawa et al., 2005*). Evolutionarily, ASK1 is a conserved protein expressed by many species ranging from *C. elegans* and drosophila to mammals (*Ausubel, 2005*; *Irazoqui et al., 2010*). ASK1 homologs in *C. elegans* (NSY-1) and drosophila (DASK1) are important in control of apoptosis or innate immunity (*Ausubel, 2005*; *Irazoqui et al., 2010*; *Shiizaki et al., 2013*). So ASK1-mediated signaling pathway may be conserved during evolution and be of great significance in control of innate response upon virus infection.

Previously we have investigated the roles of polyubiquitination in the regulation of innate signaling and innate cytokine production (*Wang et al., 2009*; *Yang et al., 2011*; *Chen et al., 2013*; *Xia et al., 2013*; *Liu et al., 2014*). To further explore roles of Ub modification in regulation of innate response, we screened Fbxo proteins in macrophages and identified Fbxo21 as an important regulator in antiviral innate immunity. Since roles of ASK1 in TLR response has been reported (*Matsuzawa et al., 2005*) while ASK1 is identified as the potential target of Fbxo21 in current study, we focused on the roles of Fbxo21 in antiviral response. We found that Fbxo21, by joining with Skp1-Cul1-Rbx1 (SCF^Fbxo21), is required for the production of inflammatory cytokines and type I IFNs upon vesicular stomatitis virus (VSV) and herpes simplex virus 1 (HSV-1) infection. Our study suggests that Fbxo21 facilitates non-proteolytic Ub-modification of ASK1, adding insights into the mechanistic understanding of antiviral innate response.

## Results

### Identification of Fbxo21 in innate response of macrophages

By using RAW264.7 cells treated with VSV and HSV-1, we screened the expression of Fbxo proteins by using GeneChip Microarrays (*Figure 1A*). We then searched Pubmed of NCBI for function-unknown Fbxo members. Taking into consideration of the BioGPS database for Fbxo molecules that demonstrated characteristics of high abundance and PAMP responsiveness, we selected Fbxo8, Fbxo16, Fbxo18, Fbxo21, Fbxo38, Fbxo41, Fbxo42 and Fbxo46 as potential Fbxo candidates involved in innate immunity (*Figure 1—figure supplement 1A*). To confirm their functions in innate response, we performed Q-PCR assays of *Ifnb* expression in peritoneal macrophages (PMs) with knockdown of indicated Fbxo molecules (*Figure 1—figure supplement 1B*) and treated the cells with lipopolysaccharide (LPS, TLR4 agonist) and VSV. We found that Fbxo21 knockdown most significantly inhibited the mRNA levels of *Ifnb* upon LPS or VSV treatments (*Figure 1—figure supplement 1C,D*). Moreover, we found that Fbxo21 is abundantly expressed by multiple tissues and macrophages (*Figure 1—figure supplement 1E*). These data suggest that Fbxo21 is a potential candidate of Fbxo members involved in innate immunity.

### Fbxo21 is required for antiviral innate response

We then analyzed the effects of Fbxo21 knockdown (*Figure 1B*) on innate immune response in detail. We found that upon LPS treatments, the production of IL-6 and IFNβ by PMs, bone marrow-derived macrophages (BMMs) and RAW264.7 cells was all significantly decreased; But Fbxo21 knockdown only minimally affected IL-6 and IFNβ production of macrophages treated with CpG-ODN (TLR9 agonist) and TLR3 agonist poly (I:C) (*Figure 1—figure supplement 2A–F*). Upon infection with VSV and HSV-1, the production of IL-6 and IFNβ was significantly impaired in macrophages after Fbxo21 knockdown (*Figure 1C–F*). These data suggest that Fbxo21 is required for innate antiviral response in macrophages.

We further validated the roles of Fbxo21 in antiviral response by depleting Fbxo21 expression using CRISPR-Cas9–mediated genome editing technique in RAW264.7 cells and L929 fibroblasts (*Figure 2A*). *Fbxo21^-/-* RAW264.7 cells demonstrated significantly impaired IL-6 and IFNβ production in response to LPS treatment, transfection of poly (I:C), poly (dA:dT) and poly (dG:dC), or infection with VSV and HSV-1 (*Figure 2B–E*). More importantly, Fbxo21 deficiency significantly promoted the replication of VSV and HSV-1 in RAW264.7 cells and L929 fibroblasts (*Figure 2F,G*). By using VSV expressing green fluorescent protein (VSV-GFP), we confirmed that Fbxo21 deficiency promoted the replication of VSV (*Figure 2H*). These data convincingly demonstrate that Fbxo21 is required for antiviral innate response.

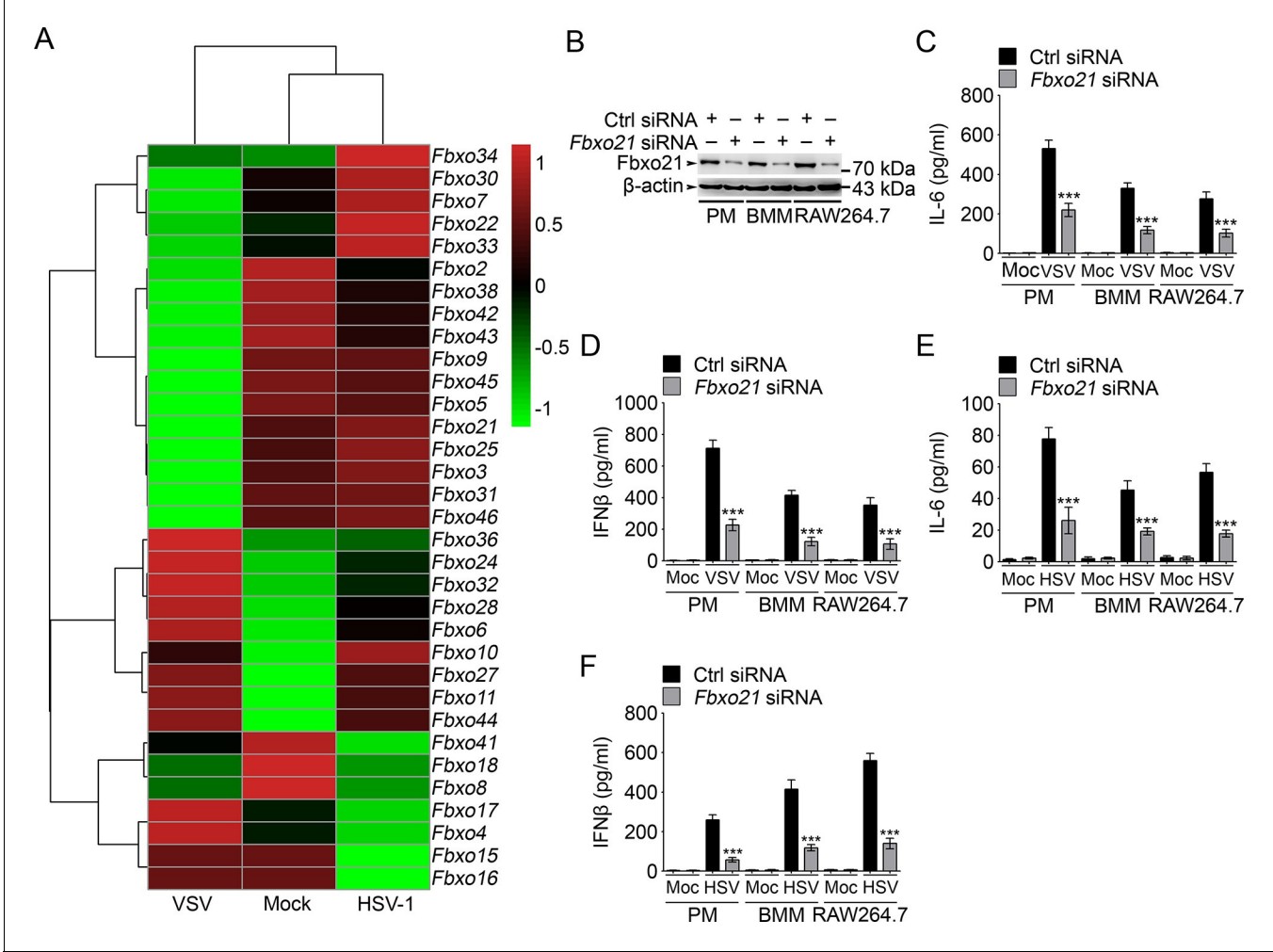

**Figure 1.** Knockdown of Fbxo21 inhibits innate antiviral response of macrophages. (**A**) Heat map representation of F-box genes in RAW264.7 cells infected with VSV (MOI = 1) or HSV-1 (MOI = 5). (**B**) Efficiency of Fbxo21 knockdown in macrophages, as examined by immunoblot 48 hr after transient transfection of indicated control (Ctrl) or *Fbxo21* siRNAs. One representative experiment of three was shown. PM, peritoneal macrophages; BMM, bone marrow-derived macrophages. (**C–F**) Cells (1 x 10$^5$ cells per 24-well) in (**B**) were infected with or without (Moc) VSV (**C, D**; MOI = 1) or HSV-1 (**E, F**; MOI = 5) as indicated for 8 hr. IL-6 (**C, E**) and IFNβ (**D, F**) concentrations in supernatant were determined by ELISA. Error bars indicated for mean ± SD of triplicate samples. Data were analyzed by one-way ANOVA followed by Bonferroni multiple comparison using PRISM software (\*\*\*p < 0.001; versus control siRNA group infected with VSV or HSV-1).

The following figure supplements are available for figure 1:

**Figure supplement 1.** Identification of Fbxo21 as an important regulator of innate response.

**Figure supplement 2.** Knockdown of *Fbxo21* impairs innate TLR4 response.

## Fbxo21 promotes the activation of IRF3 and AP-1 pathways

In *Fbxo21$^{-/-}$* RAW264.7 cells, we transiently overexpressed Fbxo21 (*Figure 3A*). We found that Fbxo21 dose-dependently increased the mRNA expression of *Il6* and *Ifnb* induced by VSV and HSV-1 (*Figure 3B–E*). Correspondingly, we found that Fbxo21 overexpression promoted the transactivation of AP-1, IFNβ and IRF3 reporters, but not NF-κB reporter, upon VSV and HSV-1 infection (*Figure 3F–I*). As further evidence, the nuclear levels of active DNA-bound c-fos and IRF3, but not p65, were significantly decreased in *Fbxo21$^{-/-}$* cells (*Figure 3J–O*). Therefore Fbxo21 may promote innate antiviral response by activating the AP-1 and IRF3 signaling pathways.

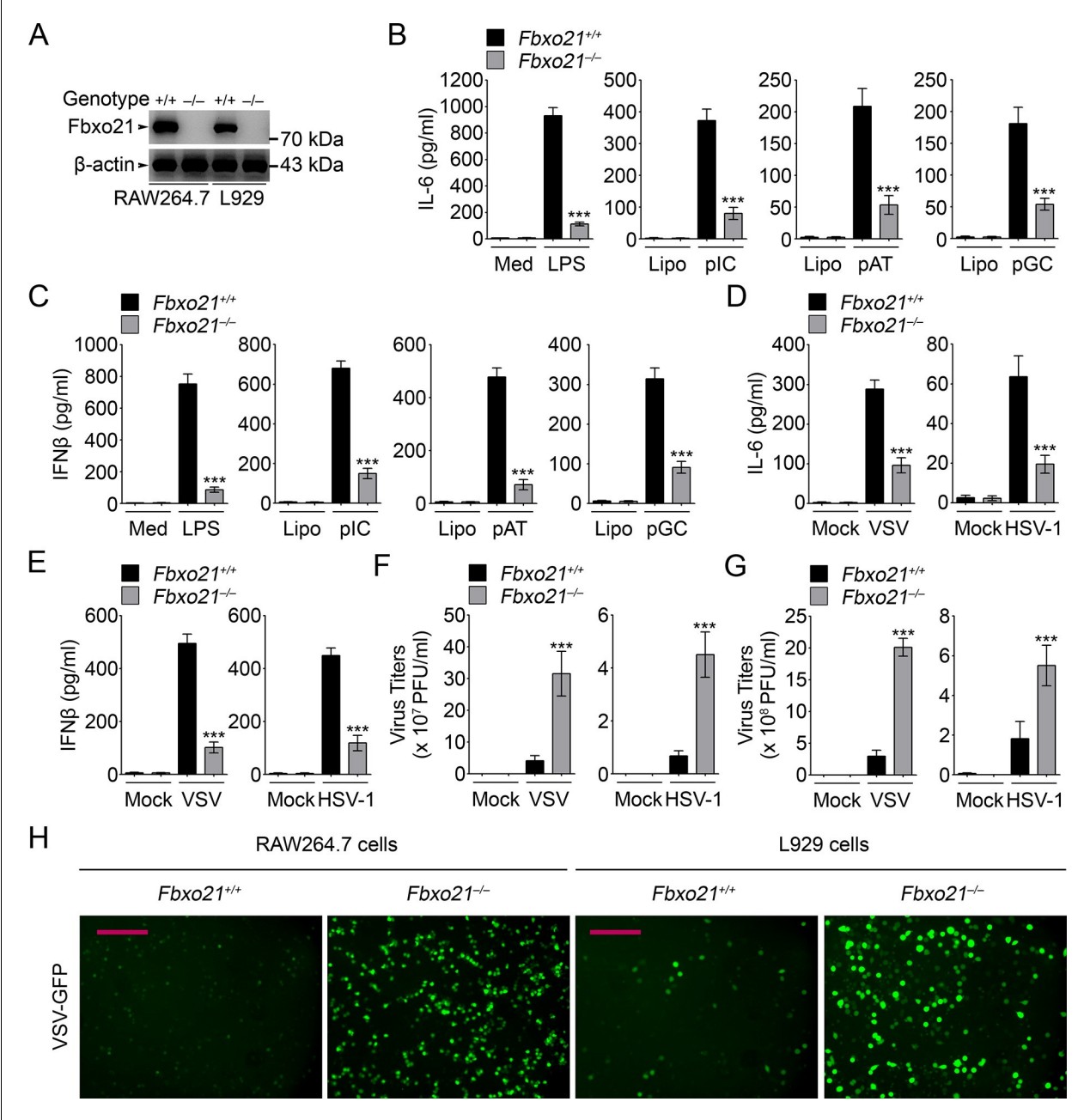

**Figure 2.** Deficiency of Fbxo21 impairs antiviral innate response. (**A**) Confirmation of Fbxo21 depletion. Fbxo21 expression in single-clone-derived RAW264.7 cells and L929 cells established by CRISPR-Cas9 editing was examined by immunoblot. One representative experiment of three was shown. (**B, C**) *Fbxo21*[+/+] and *Fbxo21*[-/-] RAW264.7 cells ($1 \times 10^5$ cells per 24-well) were treated with or without 100 ng/ml LPS (for 6 hr), or transfected with 2.5 μg/ml of liposome-packaged poly (I:C), poly (dA:dT) or poly (dG:dC) (for 8 hr). IL-6 (**B**) and IFNβ (**C**) concentrations in supernatant were determined by ELISA. Med, medium; CpG, CpG-ODN; pIC, poly (I:C). (**D–G**) *Fbxo21*[+/+] and *Fbxo21*[-/-] RAW264.7 cells (**D–F**) or L929 cells (**G**) were infected with or without (Mock) VSV (MOI = 1) or HSV-1 (MOI = 5) as indicated for 8 hr (**D, E**) or 12 hr (**F, G**). IL-6 (**D**) and IFNβ (**E**) concentrations in supernatant were determined by ELISA. The titers of viruses in cells were determined by plaque formation assay (**F, G**). In (**B–G**), error bars indicated for mean ± SD of triplicate samples. Data were analyzed by one-way ANOVA followed by Bonferroni multiple comparison using PRISM software (ns, not significant; ***p < 0.001; versus corresponding *Fbxo21*[+/+] group). (**H**) *Fbxo21*[+/+] and *Fbxo21*[-/-] RAW264.7 cells were infected with or without (Mock) VSV-GFP virus (MOI = 1) for 8 hr. Then cells were viewed under immunofluorescence microscope. One representative experiment of three was shown. Scale bars: 200 μm.

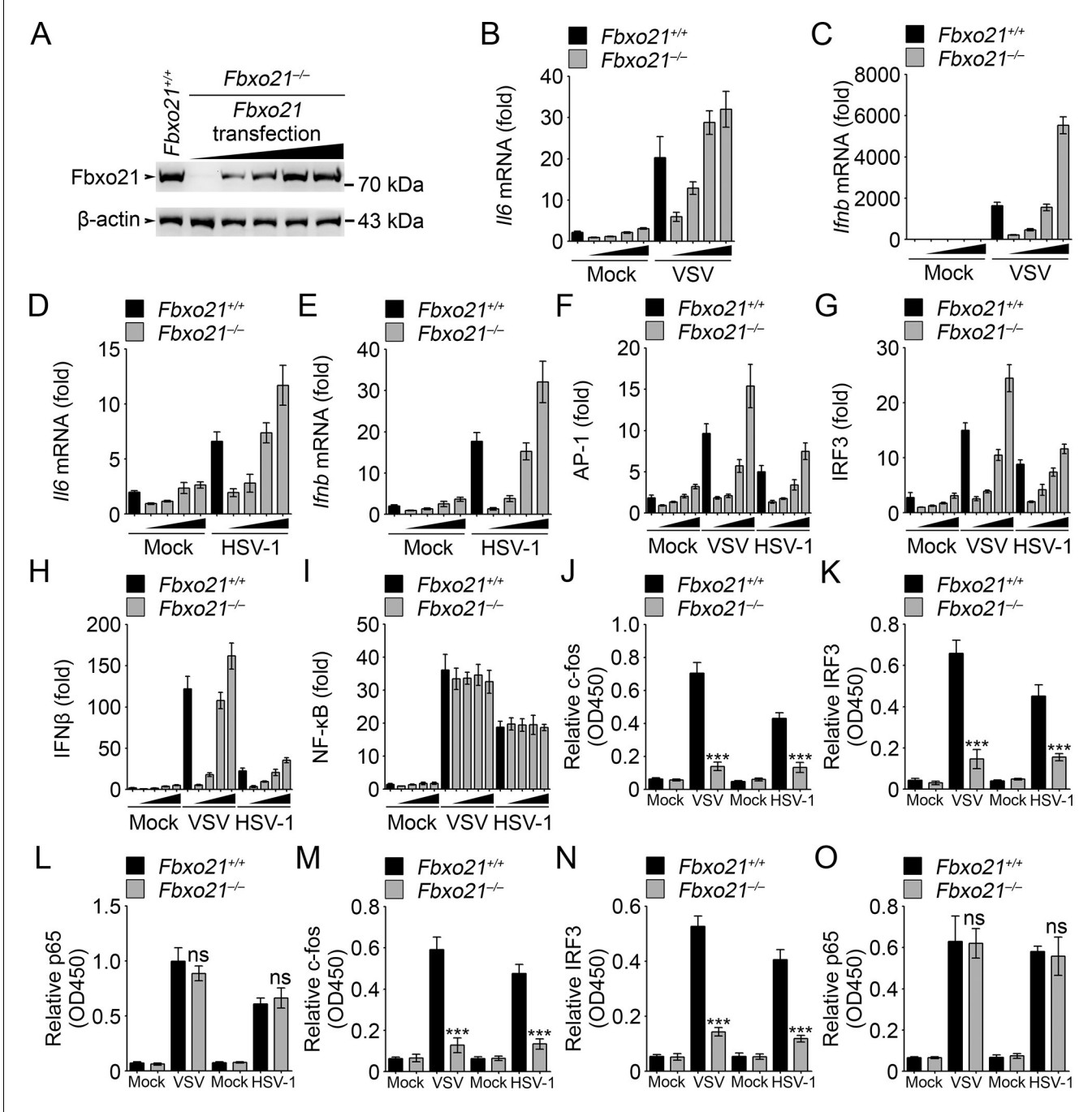

**Figure 3.** Fbxo21 overexpression promotes innate antiviral response, and activates AP-1 and IRF3-IFNβ signaling pathway. (**A**) *Fbxo21*−/− RAW264.7 cells were transfected with Fbxo21 vectors (0, 0.1, 0.2, 0.5 and 1 μg respectively) for 48 hr. Fbxo21 expression was evaluated by immunoblot. (**B–E**) Cells in (A, transfected with 0, 0.1, 0.2 and 0.5 μg Fbxo21 vectors respectively) were infected with VSV (MOI = 1) or HSV-1 (MOI = 5) for 8 hr. Then indicated cytokines were examined by Q-PCR. *β-actin* was used as internal control in the Q-PCR assays. (**F–I**) Cells in (A, transfected with 0, 0.1, 0.2 and 0.5 μg Fbxo21 vectors respectively) were also transfected with indicated reporters and infected with VSV or HSV-1 for 4 hr. Then reporter transactivation was measured by dual-luciferase activity assays. (**J–O**) *Fbxo21*+/+ and *Fbxo21*-/- RAW264.7 cells (**J–L**) or L929 cells (**M–O**) were infected with VSV (MOI = 1) or HSV-1 (MOI = 5) for 4 hr. Then nuclear extracts were examined for indicated transcription factors bound to specific DNA sequence by ELISA. Error bars indicated for mean ± SD of triplicate samples. Data were analyzed by one-way ANOVA followed by Bonferroni multiple comparison using PRISM software (ns, not significant; ***p<0.001; versus corresponding *Fbxo21*+/+ group).

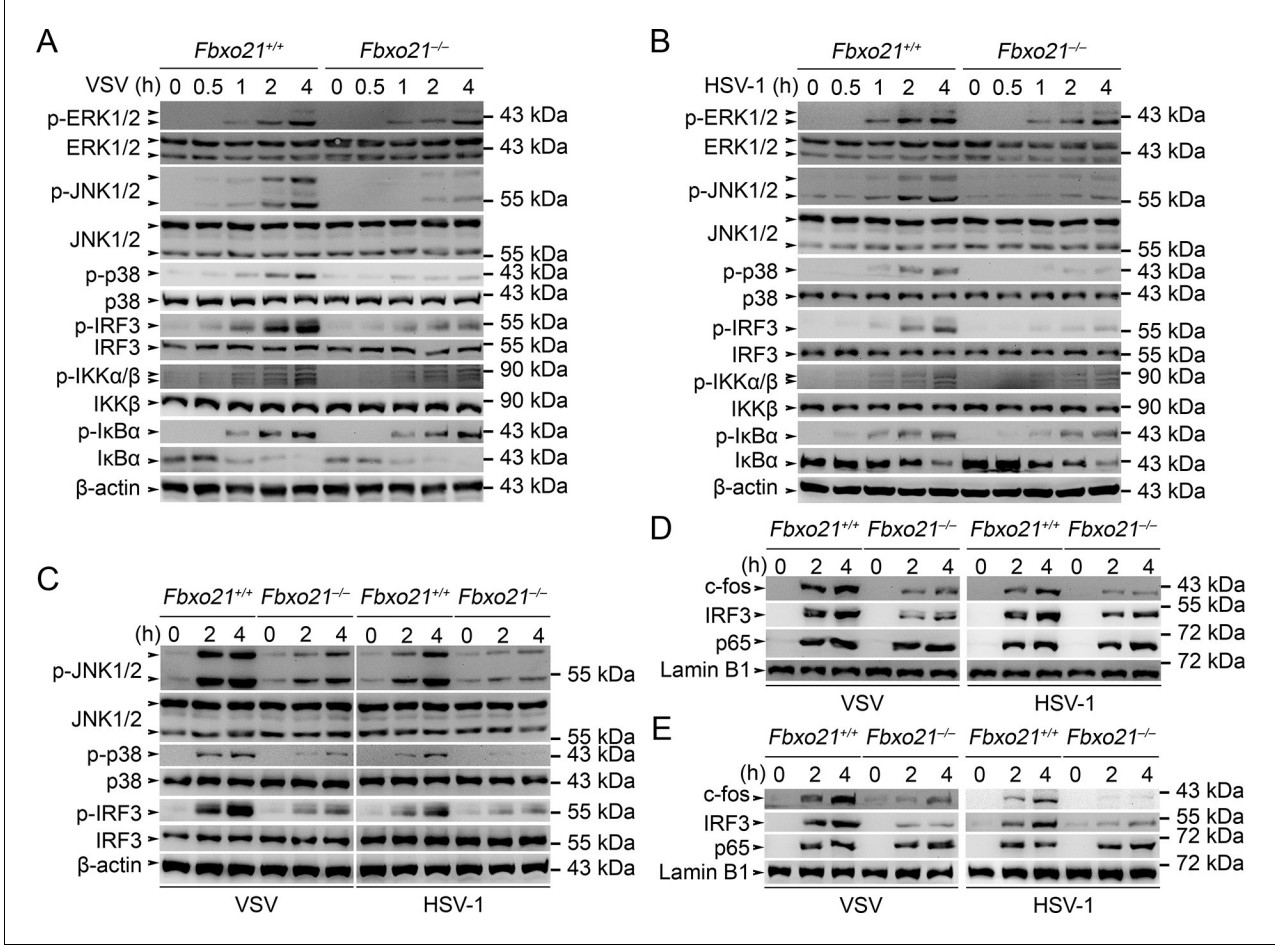

**Figure 4.** Deficiency of Fbxo21 inhibits virus-induced activation of JNK/p38-AP-1 and IRF3. (**A–C**) *Fbxo21*<sup>+/+</sup> and *Fbxo21*<sup>-/-</sup> RAW264.7 cells (**A, B**) or L929 cells (**C**) were infected with VSV (MOI = 1) or HSV-1 (MOI = 5) as indicated. The phosphorylation of indicated molecules was examined by immunblot. (**D, E**) *Fbxo21*<sup>+/+</sup> and *Fbxo21*<sup>-/-</sup> RAW264.7 cells (**D**) or L929 cells (**E**) were treated as in (**A–C**), and the nuclear proteins were extracted and examined for indicated transcription factors by immunoblot. One representative experiment of three was shown.

We then analyzed the signaling pathways in *Fbxo21*<sup>-/-</sup> RAW264.7 cells infected with VSV and HSV-1. We found that Fbxo21 deficiency inhibited the phosphorylation of JNK1/2 and p38 but not ERK1/2 mitogen-activated protein kinases (MAPKs), the type I IFNs-associated IRF3, but not the NF-κB-associated IKKα/β-IκBα (*Figure 4A,B*). In *Fbxo21*<sup>-/-</sup> L929 cells, the phosphorylation of JNK1/2, p38 and IRF3 was also decreased (*Figure 4C*). Moreover, the nuclear levels of c-fos and IRF3, but not p65, were lower in *Fbxo21*<sup>-/-</sup> RAW264.7 and L929 cells (*Figure 4D,E*). These data together indicate that Fbxo21 deficiency impairs virus-induced activation of the AP-1 and IRF3 signaling pathways.

## Fbxo21 is a component of the SCF complex and interacts with ASK1

We then went to analyze the potential target of Fbxo21 by using immunoprecipitations (IPs) of endogenous Fbxo21 in RAW264.7 cells plus liquid chromatography mass spectrometry (LC-MS) assays. The MS data indicated that Fbxo21 could interact with ASK1 (*Figure 5—figure supplement 1*). Using IP assays in RAW264.7 and L929 cells, we found that Fbxo21 did interact with ASK1 under resting state (*Figure 5A*). After VSV and HSV-1 infection, we examined the dynamic changes of Fbxo21-ASK1 interactions in RAW264.7 cells. We found that the interaction of Fbxo21-ASK1 was increased 0.5–2 hr after virus infection, and was then decreased 3–4 hr after virus infection (*Figure 5B*), indicating that the potential modulation of ASK1 by Fbxo21 mainly occurred within 0.5–2 hr after virus infection.

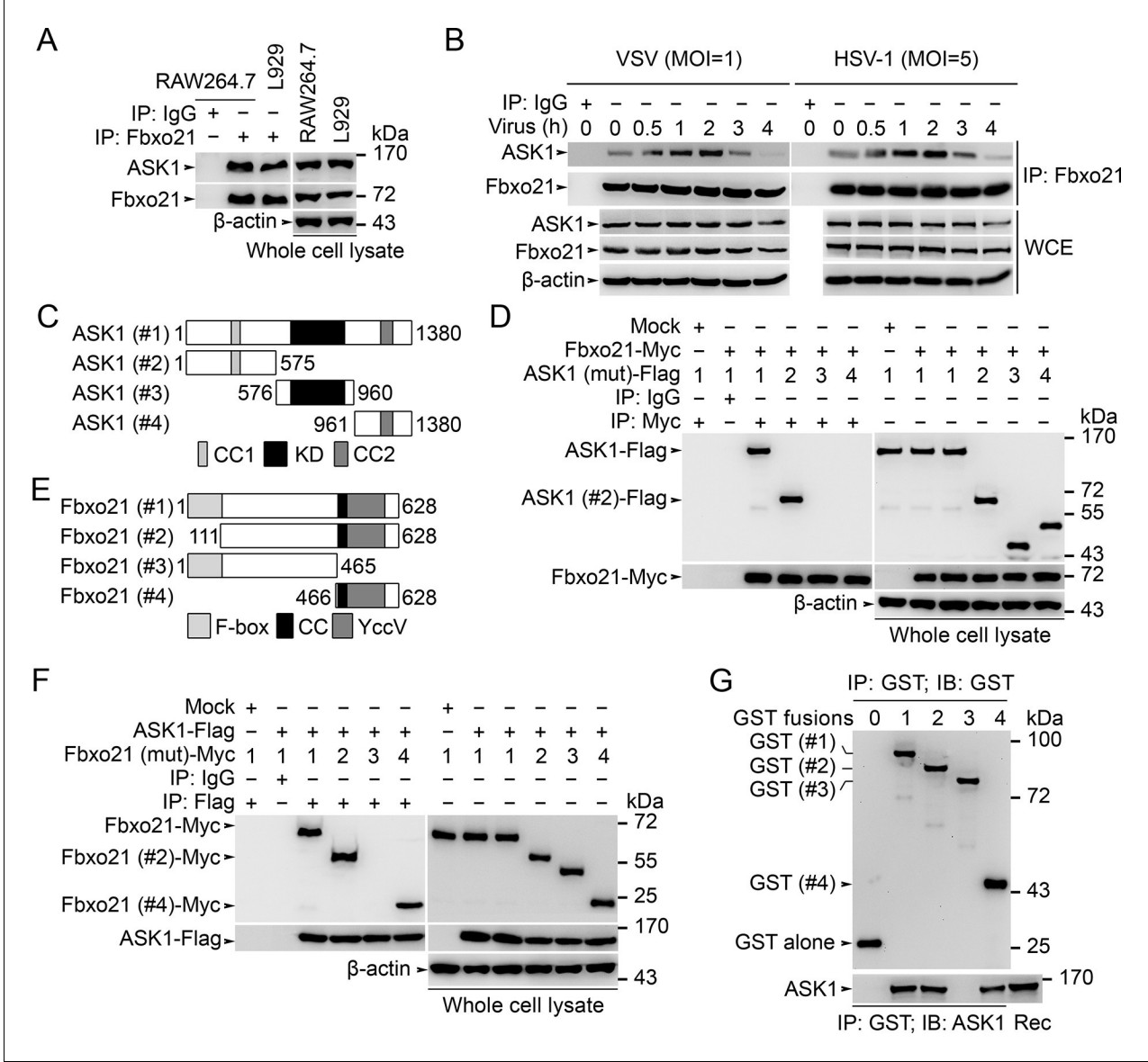

**Figure 5.** Fbxo21 interacts with ASK1. (**A, B**) Wild type RAW264.7 cells or L929 cells infected with (**B**) or without (**A**) VSV and HSV-1 were prepared for whole cell extracts (WCE), which were immunoprecipitated (IP) with anti-Fbxo21 or IgG as indicated. The associated ASK1 was examined by immunoblot. (**C**) Schema of ASK1 mutants. CC1, coiled-coil domain 1; KD, kinase domain; CC2, coiled-coil domain 2. (**D**) HEK293 cells were cotransfected with Fbxo21-Myc and indicated Flag-tagged ASK1 mutants (1–4 mutants as illustrated in (**C**)) for 48 hr. Whole cell lysates immunoprecipitated (IP) with anti-Myc agaroses were examined for Flag-tagged ASK1 (mutant) by immunoblot. (**E**) Schema of Fbxo21 mutants. F-box, F-box domain; CC, coiled-coil domain; YccV, YccV domain. (**F**) HEK293 cells were cotransfected with ASK1-Flag and indicated Myc-tagged Fbxo21 mutants (1–4 mutants as illustrated in (**E**)) for 48 hr. Whole cell lysates immunoprecipitated (IP) with anti-Flag agaroses were examined for Myc-tagged Fbxo21 (mutant) by immunoblot. (**G**) GST pull-down assays of Fbxo21 interaction with recombinant (Rec) ASK1. '0' indicates for GST alone, and 1–4 indicate for GST-fused Fbxo21 (mutant) as in (**E**). One representative experiment of three was shown.

The following figure supplements are available for figure 5:

**Figure supplement 1.** MS assays of Fbxo21-associated proteins.

**Figure supplement 2.** Fbxo21 is a component of SCF complex.

We then mapped the domains in Fbxo21 and ASK1 mediating the interaction. After transfection of Fbxo21 with ASK1 (mutants) plasmids (*Figure 5C*) in HEK293 cells, we found that ASK1 or ASK1 with only the N-terminal segment could be coimmunoprecipitated with Fbxo21 (*Figure 5D*). When ASK1 was cotransfected with Fbxo21 (mutants) (*Figure 5E*) in HEK293 cells, it was found that full-length Fbxo21 or C-terminal segment of Fbxo21 could be coimmunoprecipitated with ASK1 (*Figure 5F*). As further evidence, we performed GST pull-down assays and found that GST fusion proteins of Fbxo21 and Fbxo21 (466–628 aa) could pull down recombinant ASK1 (*Figure 5G*). These data suggest that the C-terminal part of Fbxo21 that contains one coiled-coil domain and one YccV-like protein domain, may interact with the N-terminal part of ASK1 that contains one coiled-coil domain and one thioredoxin-binding domain.

To validate that Fbxo21 is a component of the SCF complex, we examined the endogenous Skp1, Cul1 and Rbx1 in Fbxo21 immune complexes, and found that Skp1, Cul1 and Rbx1 could be coimmunoprecipitated with Fbxo21 (*Figure 5—figure supplement 2A*). Then we cotransfected Fbxo21 with conserved SCF components (Skp1, Cul1 or Rbx1) in HEK293 cells. We found that Fbxo21, but not Fbxo21 without F-box domain, could interact with Skp1, Cul1 and Rbx1 (*Figure 5—figure supplement 2B–D*). These data indicate that Fbxo21 could form a complex with Skp1, Cul1 and Rbx1 (SCF$^{Fbxo21}$).

## Fbxo21 is required for Lys29-linkage of ASK1

We then asked how SCF$^{Fbxo21}$ affected the signaling pathways via ASK1. Since SCF complexes usually mediate the Ub-modification and proteasome degradation of substrates (*Skaar et al., 2013; 2014*), we evaluated the ubiquitination and degradation of ASK1 in *Fbxo21$^{-/-}$* RAW264.7 cells. We found that ASK1 could be polyubiquitinated after VSV and HSV-1 infection, which was significantly impaired in *Fbxo21$^{-/-}$* RAW264.7 cells (*Figure 6A*). However, we found that the degradation of ASK1 in *Fbxo21$^{-/-}$* RAW264.7 cells was not affected (*Figure 6B*). These data indicate that SCF$^{Fbxo21}$ may mediate non-proteolytic polyubiquitination of ASK1.

It was noted that Fbxo21 protein was also decreased after VSV and HSV-1 infection (*Figure 6B*) while the mRNA levels of *Fbxo21* were differentially regulated by VSV and HSV-1 infection (*Figure 1A*). By using Q-PCR assays, we confirmed that *Fbxo21* mRNA was induced by HSV-1 infection but decreased by VSV infection (*Figure 6—figure supplement 1A*). To reconcile this discrepancy, we examined the protein levels of Fbxo21 in RAW264.7 cells after VSV and HSV-1 infection in the presence of a transcription inhibitor (actinomycin D, Act D) or a proteasome inhibitor (MG132). In the presence of Act D, the amounts of Fbxo21 protein were decreased after VSV and HSV-1 infection (*Figure 6—figure supplement 1B*), indicating that Fbxo21 itself may be regulated by proteasomal degradation. In the presence of MG132 that blocks the degradation of Fbxo21, we found that VSV infection could not decrease Fbxo21 while HSV-1 infection could induce the expression of Fbxo21 protein 12 hr after infection (*Figure 6—figure supplement 1C*), confirming that VSV and HSV-1 may differentially regulate the transcription of Fbxo21. Given that Fbxo21 was abundantly expressed in many cell types, it may be inferred that Fbxo21 may act at an early stage of antiviral response while at a later stage the transcriptional regulation of *Fbxo21* mRNA may contribute to the differences in innate response to RNA virus versus DNA virus.

We then analyzed the Ub-modification type of ASK1 by SCF$^{Fbxo21}$. We found that Ub with only the intact Lys29 residue (K29O-Ub) could be linked to ASK1 by Fbxo21 (*Figure 6C*). In *Fbxo21$^{-/-}$* RAW264.7 cells reconstituted with Fbxo21, we found that Fbxo21 could promote the linkage of K29O-Ub to ASK1 after virus infection (*Figure 6D,E*). When the cells were treated with MLN-4924, an inhibitor of NEDD8-activating enzyme that has been implicated in the assembly of SCF complex (*Soucy et al., 2009*), we found that Fbxo21-mediated Lys29-linkage of ASK1 was abrogated (*Figure 6D,E*), indicating that Fbxo21-mediated ASK1 modification requires the assembly of SCF$^{Fbxo21}$ complex. These data suggest that Fbxo21, by joining with Skp1-Cul1-Rbx1 (SCF$^{Fbxo21}$), is required for Lys29-linkage of ASK1.

As direct evidence, we performed in vitro polyubiquitination assays. Since the E2 enzyme(s) specific for Lys29-linkage of Ub chains have not been identified, we utilized conjugation fraction A that contains predominantly E1 and E2 enzymes instead of defined E2s in the in vitro*in vitro* polyubiquitination assays. In the presence of recombinant GST-Fbxo21, ASK1 and Cul1-Rbx1-Skp1, as well as the NEDD8 conjugation system (neddylation), ASK1 could be efficiently polyubiquitinated; while in the presence of MLN-4924, the polyubiquitination of ASK1 was significantly inhibited (left panel,

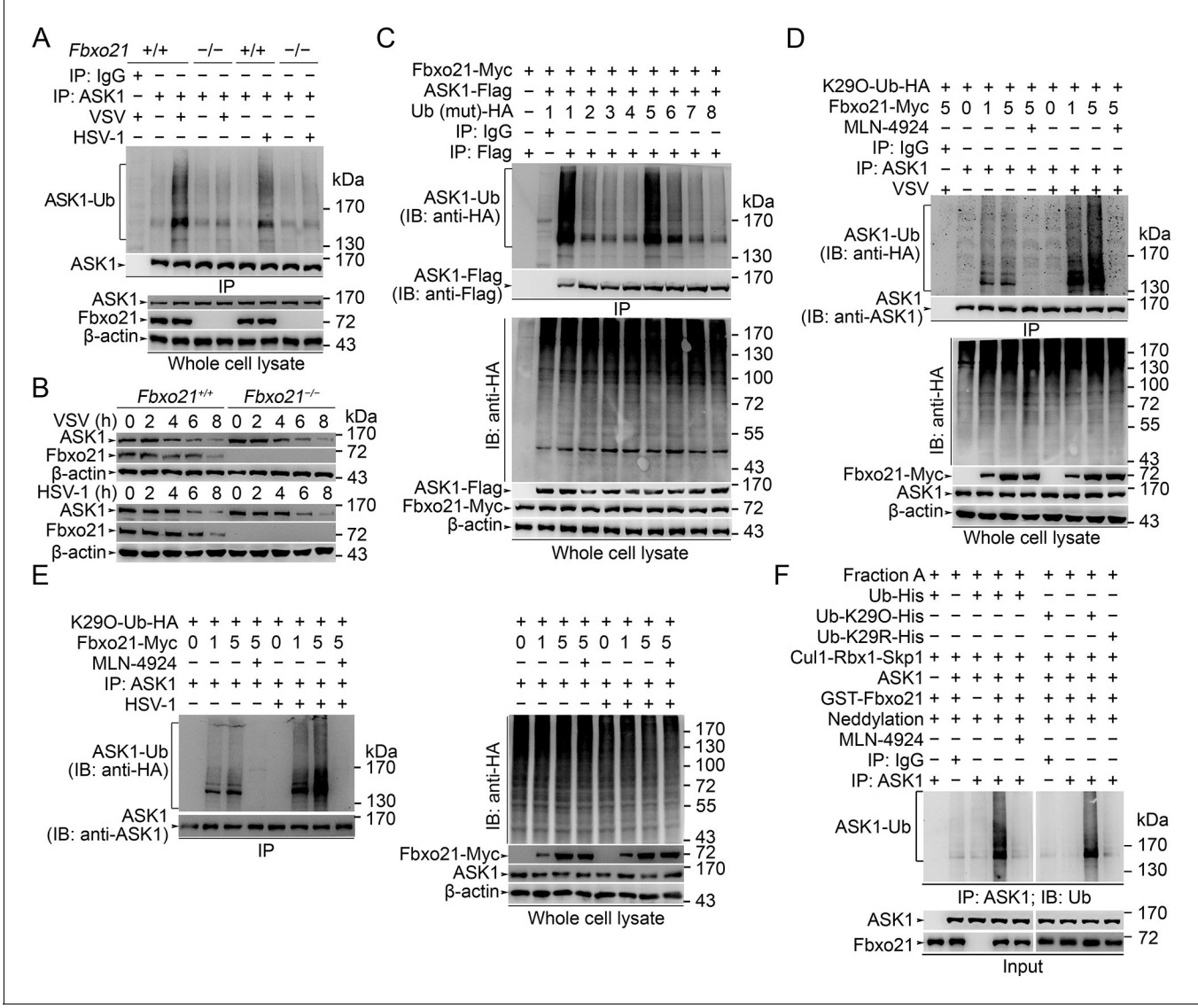

**Figure 6.** SCF[Fbxo21] mediates virus-induced Lys29-linkage of ASK1. (**A**) *Fbxo21*[+/+] and *Fbxo21*[-/-] RAW264.7 cells were infected with VSV (MOI=1) or HSV-1 (MOI = 5) for 2 hr. Then polyubiquitination of ASK1 was examined by immunoblot (IB) after immunoprecipitations (IP). (**B**) *Fbxo21*[+/+] and *Fbxo21*[-/-] RAW264.7 cells were infected with VSV (MOI = 1) or HSV-1 (MOI = 5) as indicated. ASK1 and Fbxo21 levels in whole cell lysates were examined by immunoblot. (**C**) Myc-tagged Fbxo21 and Flag-tagged ASK1 were cotransfected with HA-tagged Ub (mutant) into HEK293 cells for 48 hr, and infected with VSV (MOI = 1) for 2 hr. Then polyubiquitinated ASK1 was examined by immunoblot (IB) against HA after immuneprecipitations (IP) with anti-Flag agaroses. 1, wild type Ub; 2, Ub with only the Lys6 residue unchanged (K6O-Ub); 3, K11O-Ub; 4, K27O-Ub; 5, K29O-Ub; 6, K33O-Ub; 7, K48O-Ub; 8, K63O-Ub. (**D, E**) *Fbxo21*[-/-] RAW264.7 cells were transfected with indicated amounts (0, 1 or 5 μg) Fbxo21-Myc and equal amounts of K29O-Ub-HA vectors for 48 hr, and then infected with VSV (MOI = 1) or HSV-1 (MOI = 5) for 2 hr in the presence or absence of MLN-4924 (10 nM). Then polyubiquitinated ASK1 was examined by immunoblot (IB) against HA after immuneprecipitations (IP). (**F**) After incubation for 30 min, the in vitro polyubiquitination system was boiled for 5 min, and then polyubiquitinated ASK1 was examined by immunoblot (IB) against Ub after immuneprecipitations (IP). One representative experiment of three was shown.

The following figure supplement is available for figure 6:

**Figure supplement 1.** Analysis of Fbxo21 expression and degradation.

*Figure 6F*). To confirm the Lys29-linkage of Ub chains to ASK1, we used 6xHis-Ub-K29O (with only Lys29 in Ub) and 6xHis-Ub-K29R (with the Lys29 mutated into Arg in Ub) in the in vitro assays, and found that the former, but not the later, could be linked to ASK1 (right panel, *Figure 6F*). These data further validate that SCF[Fbxo21] could promote the Lys29-linkage of ASK1.

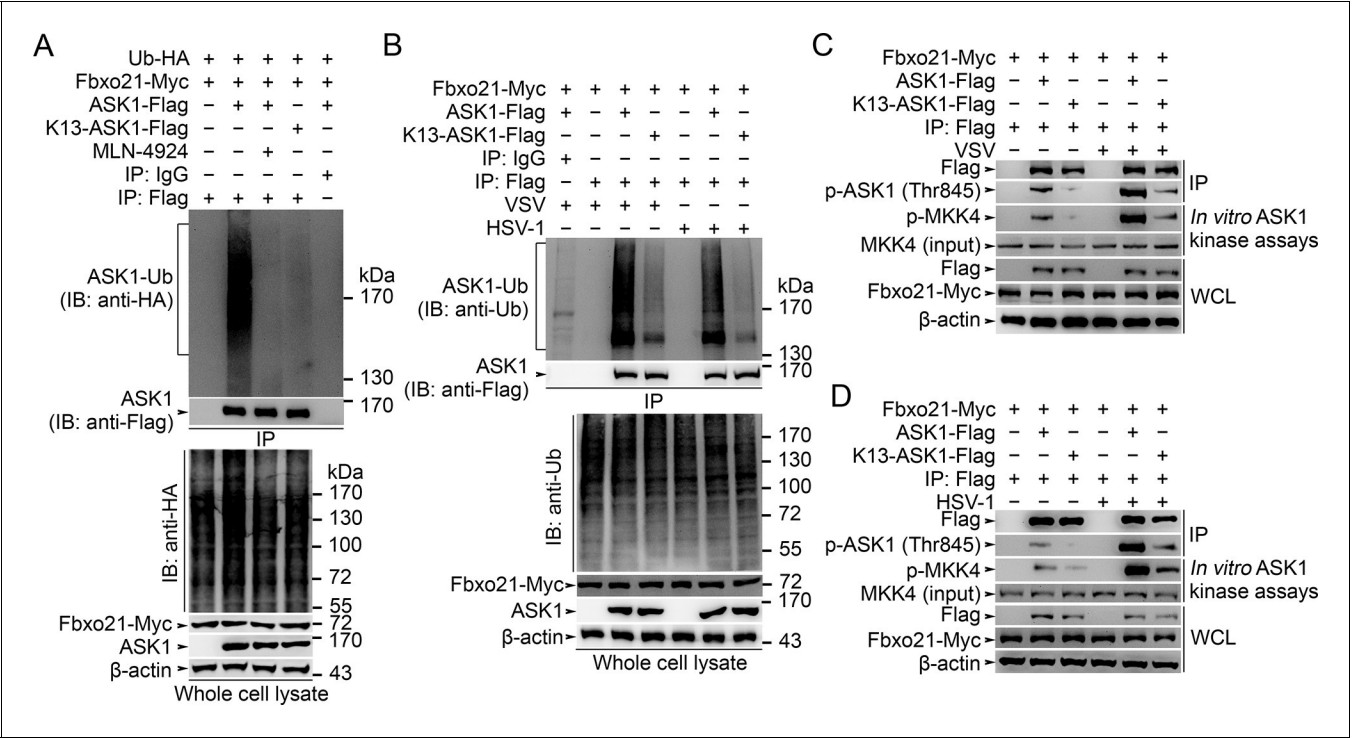

**Figure 7.** Lys29-linkage of ASK1 is crucial for Fbxo21-mediated ASK1 activation. (**A**) HEK293 cells transiently transfected with indicated vectors for 48 hr and infected with VSV (MOI = 1) for 2 hr in the presence or absence of MLN-4924 (10 nM). Then ASK1 (mutant) was immunoprecipitated (IP) with anti-Flag agaroses, and polyubiquitination was evaluated by immunblot (IB) against HA. (**B**) *Fbxo21⁻/⁻* RAW264.7 cells were transfected with indicated vectors for 48 hr and infected with VSV (MOI = 1) or HSV-1 (MOI = 5) for 2 hr. Then ASK1 (mutant) was immunoprecipitated (IP) with anti-Flag agaroses, and polyubiquitination was evaluated by immunblot (IB) against Ub. (**C, D**) *Fbxo21⁻/⁻* RAW264.7 cells were transfected with indicated vectors for 48 hr and infected with VSV (MOI = 1, C) or HSV-1 (MOI = 5, D) for 2 hr. Then ASK1 (mutant) was immunoprecipitated (IP) with anti-Flag agaroses, and the phosphorylated ASK1 was examined by immunoblot. Otherwise, the ASK1 (mutant) after IP was incubated with recombinant MKK4, and then immunoblotted for phosphorylated MKK4. One representative experiment of three was shown.

The following figure supplements are available for figure 7:

**Figure supplement 1.** Prediction of the Lys residue in ASK1 responsible for linkage with Ub chain.

**Figure supplement 2.** Confirmation of the Lys residue in ASK1 responsible for linkage with Ub chain.

**Figure supplement 3.** Fbxo21 deficiency impairs poly-Ub modification of ASK1 and then ASK1 activation.

## Lys29-linkage of ASK1 by SCF^Fbxo21 is required for ASK1 activation

We then went to identify the receptor sites for Lys29-linked Ub chain in ASK1. We predicted the ubiquitination sites in ASK1 by using the Bayesian discriminant method-prediction of ubiquitination sites algorithm (***Figure 7—figure supplement 1A***). We also performed multiple alignment of ASK1 protein sequences derived from various species (for evolution conservation; ***Figure 7—figure supplement 1B—J***). By these 2 rounds of selection, ASK1 was predicted to contain 29 ubiquitination sites. To verify these ubiquitination sites, we constructed 18 mutants for ASK1 (K1-K18, mutated indicated Lys residue(s) into Arg; ***Figure 7—figure supplement 2A***), and examined the ubiquitin levels associated with ASK1 after VSV treatments in HEK293 cells. We found that the K13 mutant of ASK1 (K13-ASK1; mutation of Lys946, Lys950, Lys951, Lys952, Lys953 and Lys957 into Arg) could most significantly decrease the levels of polyubiquitinated ASK1 (***Figure 7—figure supplement 2B*** and ***Figure 7A***). In *Fbxo21⁻/⁻* RAW264.7 cells reconstituted with Fbxo21 and infected with VSV or HSV-1, we found that wild type ASK1 but not K13-ASK1 could be polyubiquitinated (***Figure 7B***). Meanwhile, we found that K13-ASK1 demonstrated remarkably lower activity as compared to wild type ASK1 (***Figure 7C, D***). These data suggest that the K13 mutant-covered region in ASK1 may be the major

ubiquitination sites modified by Fbxo21 during virus infection, and Fbxo21-mediated Lys29-linkage of ASK1 may be required for ASK1 activation.

To further examine the contribution of SCF$^{Fbxo21}$ to ASK1 activation during antiviral innate response, we examined the phosphorylation and activity of ASK1 in *Fbxo21$^{-/-}$* cells and in MLN-4924-treated cells. We found that the phosphorylation and activity of ASK1 were significantly impaired in both *Fbxo21$^{-/-}$* cells and MLN-4924-treated cells after VSV or HSV-1 infection (*Figure 7—figure supplement 3A*). These data suggest that SCF$^{Fbxo21}$ is crucial for the phosphorylation and kinase activity of ASK1.

However, the relationship of ASK1 polyubiquitination with ASK1 activation has not been clearly demonstrated. To determine the potential roles of ASK1 polyubiquitination in ASK1 activation, we examined the dynamics of ASK1 modifications after VSV infection in *Fbxo21$^{+/+}$* and *Fbxo21$^{-/-}$* RAW264.7 cells. We found that Fbxo21 deficiency significantly decreased the poly-Ub levels of ASK1 (*Figure 7—figure supplement 3B*). However, the K48- and K63-linked poly-Ub levels of ASK1 were not significantly affected. Moreover, the phosphorylation of ASK1 as well as the kinase activity of ASK1 both occurred later than poly-Ub modification of ASK1 (*Figure 7—figure supplement 3C*). These data together indicate that SCF$^{Fbxo21}$-mediated Lys29-linkage of ASK1 may prejudge the activation of ASK1.

## Lys29-linkage of ASK1 is crucial for ASK1 and IRF3 activation

Previously ASK1 has been implicated in the regulation of TLR4-triggerred inflammatory cytokine production (*Matsuzawa et al., 2005*). However, the roles of ASK1 in regulating innate antiviral response and type I IFN production have not been fully elucidated. So we established RAW264.7 cells deficient for ASK1 (*Map3k5$^{-/-}$*) by CRISPR-Cas9 genome editing (*Figure 8A*). In *Map3k5$^{-/-}$* RAW264.7 cells, the production of IL-6 and IFNβ triggered by VSV and HSV-1 infection was significantly inhibited while the virus titers of VSV and HSV-1 were significantly increased (*Figure 8B–D*). Meanwhile, we found that the activation of JNK1/2, p38 and IRF3 was significantly impaired in *Map3k5$^{-/-}$* RAW264.7 cells infected with VSV and HSV-1 (*Figure 8E*). When the *Map3k5$^{-/-}$* RAW264.7 cells were rescued by overexpression of ASK1, K13-ASK1 or ASK1-K716A (Lys716 mutated into Ala, kinase activity inactive), we found that ASK1, but not K13-ASK1 or ASK1-K716A, could rescue the impaired IL-6 and IFNβ production and the inhibited antiviral response (*Figure 8A–D*). Furthermore, we found that K13-ASK1 could not rescue the impairment in activation of JNK1/2, p38 and IRF3, phosphorylation on Thr845 of ASK1, and the kinase activity of ASK1 (*Figure 8F,G*). These data convincingly demonstrate that Lys29-linkage of ASK1 is required for ASK1 activation and the subsequent innate antiviral responses.

## JNK1/2 activation is crucial for Fbxo21-mediated antiviral innate response

The above results have established an essential role of SCF$^{Fbxo21}$ in ASK1-dependent antiviral innate responses. However, the downstream signaling pathways responsible for ASK1-mediated effects during VSV or HSV-1 infection have not been elucidated. Since ASK1 is a kinase mainly involved in activation of JNK1/2 and p38 (*Ichijo et al., 1997*), we tested the effects of JNK1/2 and p38 inhibitors in Fbxo21-mediated signaling pathway. We found that JNK1/2 inhibitor SP600125 significantly, and pan-p38 inhibitor SB203580 to a much lesser extent, reversed Fbxo21-mediated activation of IRF3 (*Figure 9A,B*) and the production of IL-6 and IFNβ (*Figure 9C,D*) after VSV or HSV-1 infection. These data suggest that Fbxo21-mediated antiviral innate response may mainly require ASK1-dependent activation of JNK1/2 and, to a lesser extent, p38 (*Figure 9—figure supplement 1*).

## Discussion

By initiating PRRs-triggered signaling pathway, host cells can mount the production of proinflammatory cytokines as well as type I IFNs, leading to inflammatory responses and the elimination of invading pathogens (*Gürtler and Bowie, 2013*; *Yin et al., 2015*). Our study shows that ASK1 activation facilitated by SCF$^{Fbxo21}$ ubiquitin ligase complex plays an important role in promoting virus-triggered innate response, serving as a critical module in antiviral innate response. More importantly, our study reveals a new type of Ub-modification (Lys29-linkage) in regulating kinase activation, and suggests that SCF complexes may not only mediate degradation of protein substrates but non-

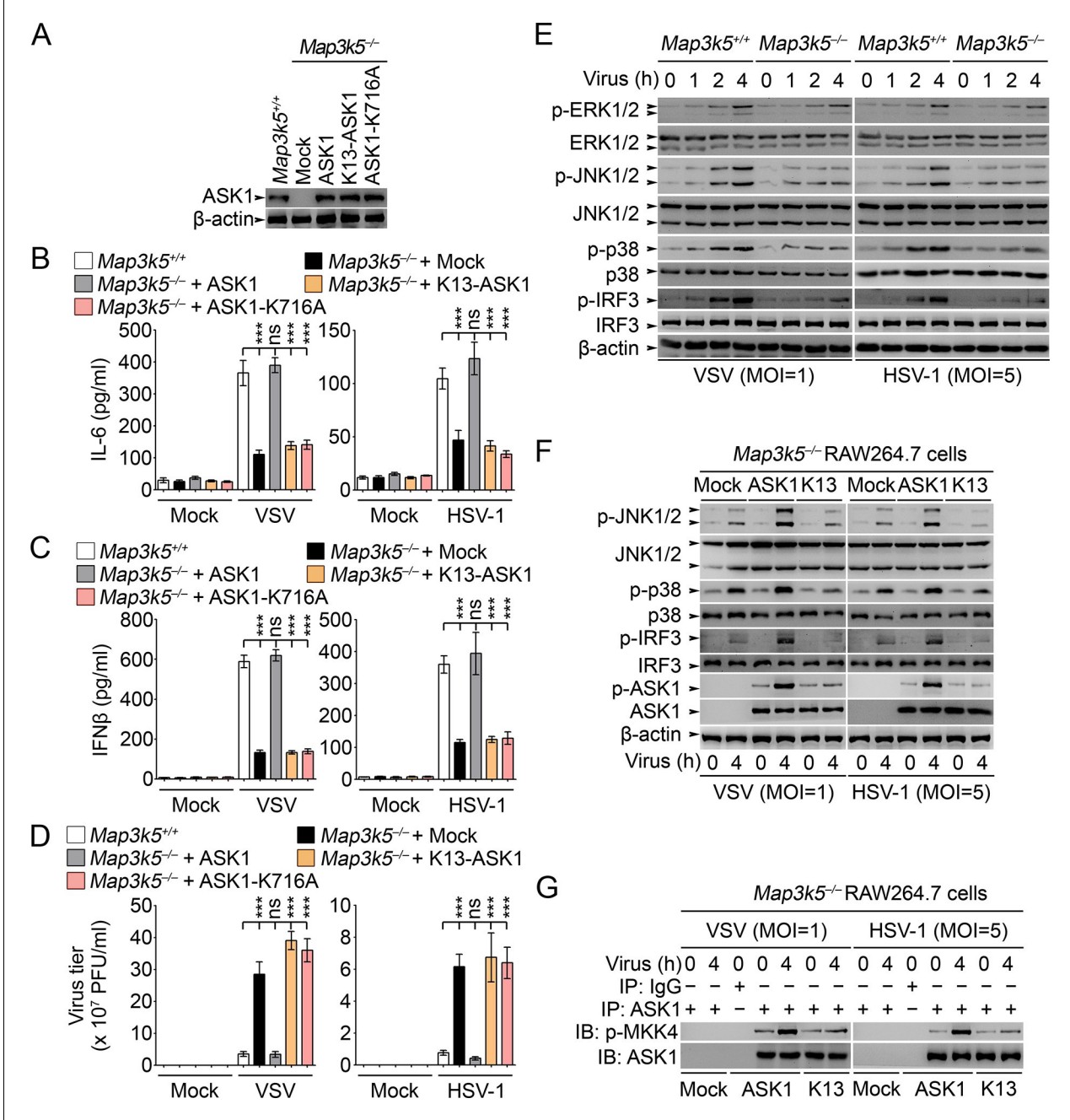

**Figure 8.** Lys29-linkage of ASK1 is crucial for innate antiviral response. (A) *Map3k5*<sup>+/+</sup> RAW264.7 cells and *Map3k5*<sup>−/−</sup> RAW264.7 cells transfected with mock vector or indicated ASK1 vectors for 48 hr were examined for ASK1 expression by Western blot. (B–D) *Map3k5*<sup>+/+</sup> and *Map3k5*<sup>−/−</sup> RAW264.7 cells (as in A) were infected with or without (Mock) VSV (MOI = 1) or HSV-1 (MOI = 5) as indicated for 8 hr (B, C) or 12 hr (D). IL-6 (B) and IFNβ (C) concentrations in supernatant were determined by ELISA. The titers of viruses in cells were determined by plaque formation assay (D). Error bars indicated for mean ± SD of triplicate samples. Data were analyzed by one-way ANOVA followed by Bonferroni multiple comparison using PRISM software (ns, not significant; ***p < 0.001). (EG) Cells in (A) were infected with or without (Mock) VSV (MOI = 1) or HSV-1 (MOI = 5) as indicated. Then whole cell lysate (E, F) or immunoprecipitates (G) were examined by Western blot. In (G), the immunoprecipitated ASK1 was examined for the kinase activity in the presence of 10 ng recombinant MKK4. One representative experiment of three was shown.

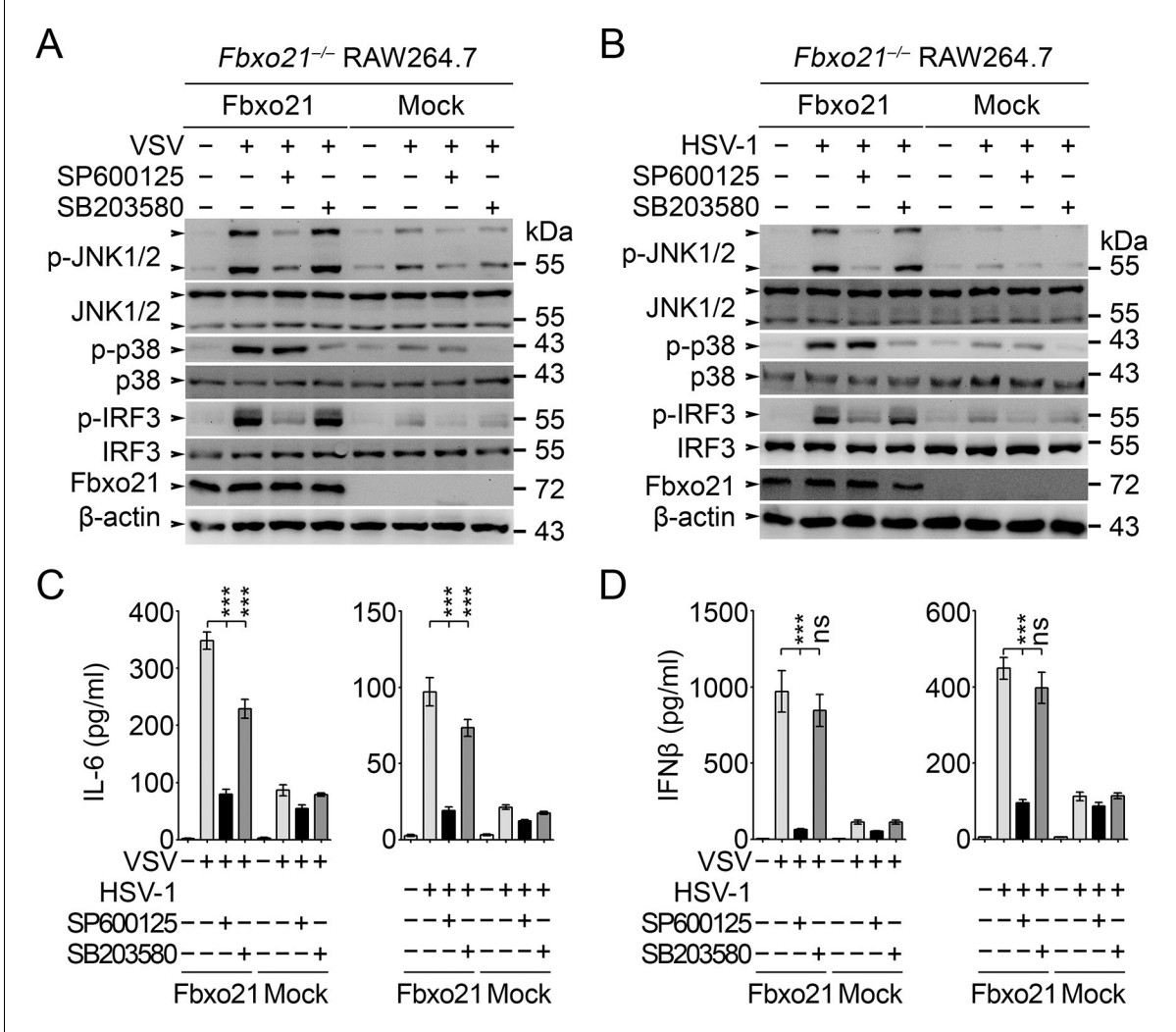

**Figure 9.** JNK1/2 activation is crucial for Fbxo21-mediated innate response. *Fbxo21⁻/⁻* RAW264.7 cells transfected with mock or Fbxo21 vectors were infected with VSV (MOI = 1) or HSV-1 (MOI = 5) for 4 hr (**A, B**) or 8 hr (**C, D**) in the presence or absence of SP600125 (20 µM) or SB203580 (20 µM). The indicated proteins were examined by immunoblot (**A, B**), and the cytokine concentrations in the supernatant were measured by ELISA (**C, D**). One representative experiment of three was shown (**A, B**). Error bars indicated for mean ± SD of triplicate samples (**C, D**). Data were analyzed by one-way ANOVA followed by Bonferroni multiple comparison using PRISM software (ns, not significant; ***p < 0.001).

The following figure supplement is available for figure 9:

**Figure supplement 1.** Proposed working model for Fbxo21-mediated innate antiviral response.

proteolytic regulation of substrates. Considering that ASK1 is a remarkably conserved kinase crucial in defense of infection (*Ausubel, 2005*; *Irazoqui et al., 2010*), our study may be of great significance for understanding of innate response in species other than mammals. However, in vivo roles of Fbxo21 may need to be confirmed by using Fbxo21 knockout mice.

Polyubiquitination of protein substrates represents a key posttranslational modification. Many single subunit E3 ubiquitin ligases have been implicated in regulating innate immune response (*Jiang and Chen, 2011*; *Zinngrebe et al., 2014*). However, little is known about the roles of the multisubunit ubiquitin ligases in innate immunity. In SCF complex, the recognition of substrate is determined by F-box proteins (*Skaar et al., 2013*; *2014*). Our study identifies Fbxo21 as a component of SCF complex and interacts with ASK1 to regulate the ubiquitination by SCF complex, thus expanding the family of SCF complex in immunity. Previously many SCF complexes have been reported whereas few of them are involved in innate immune response (*Skaar et al., 2013*; *2014*). The best

examples may be SCF$^{Fbxw1}$ (β-Trcp1) and SCF$^{Fbxw11}$ (β-Trcp2) that mediate the processing of NF-κB components and degradation of IκB isoforms (*Yaron et al., 1998*; *Shirane et al., 1999*; *Spencer et al., 1999*; *Tan et al., 1999*; *Wu and Ghosh, 1999*; *Orian et al., 2000*; *Fong and Sun, 2002*; *Lang et al., 2003*; *Amir et al., 2004*). The SCF$^{FBXW7α}$ has also been implicated in processing p100 subunit of NF-κB (*Busino et al., 2012*; *Fukushima et al., 2012*). Fbxo3 has been implicated in the degradation of Fbxl2, leading to impaired degradation of TRAF proteins and exaggerated inflammatory response (*Chen et al., 2013*). Interestingly, *GogB* from *Salmonella* can interact with human Fbxo22 and inhibit inflammatory response (*Pilar et al., 2012*). Our study suggests that Fbxo21 is required for innate antiviral response by activating ASK1-JNK signaling pathway, thus revealing an important role of Fbxo21 in innate immunity. More importantly, we have identified a new modification of ASK1, that is, Lys29-linkage of ASK1. Ub-modification of ASK1 has been reported previously. E3 ligases, such as cIAP-1, SOCS1, zinc-finger protein A20 and Stub1 (also known as CHIP), have been implicated in degradation of ASK1 and thus a negative regulator of JNK and/or p38 activation and apoptosis (*Zhao et al., 2007*; *Yu et al., 2009*; *Zhang et al., 2009*; *Won et al., 2010*). In all these studies, Ub-modifications lead to ASK1 degradation. Since most of SCF complexes mediate degradation of substrates (*Skaar et al., 2013*; *2014*), our finding of SCF$^{Fbxo21}$-mediated non-proteolytic polyubiquitination of ASK1 may indicate that SCF complexes may exert functions via non-proteolytic modification of substrates.

Ubiquitin serves as a small molecular marker on proteins (*Jiang and Chen, 2011*; *Zinngrebe et al., 2014*). Ideally, the seven lysine residues of Ub (lys6, lys11, lys27, lys29, Lys33, lys48 and Lys63) can all be transferred to protein substrates by the ubiquitin system. Lys48-linkage and Lys63-linkage of Ub have been most extensively investigated. Similar to the effects of Lys63-linkage, Lys33-linkage of CD3ζ promotes its phosphorylation and interaction with Zap70 and possibly the transport of CD3ζ to lysosomes for degradation (*Ouchida et al., 2008*; *Huang et al., 2010*). Recently our study revealed that K33-linkage of Zap70 by Nrdp1 promoted the dephosphorylation of Zap70 and thus terminated early TCR signaling in CD8$^+$ T cells (*Yang et al., 2015*). For the other types of Ub-modifications in innate response, little is known. TRAF6 has been implicated in Lys29-linkage but aggregation of huntingtin protein (*Zucchelli et al., 2011*). The E3 ligase Itch may mediate Lys29-linked polyubiquitination and degradation of Notch (*Chastagner et al., 2008*). Previously deubiquitinating enzyme USP9X is implicated in the removal of Lys29-linked ubiquitin from AMPK-related kinases (*Al-Hakim et al., 2008*). Meanwhile, USP9X stabilizes ASK1 by removal of Ub chains from ASK1 and prevention of ASK1 degradation (*Nagai et al., 2009*). Our study suggests that SCF$^{Fbxo21}$ may be the ligase responsible for Lys29-linkage of ASK1 by demonstrating that Fbxo21 is required for Lys29-linkage of ASK1 upon viral infection, leading to non-proteolytic modification of ASK1 and ASK1 phosphorylation. Our data also indicate that Lys29-linkage of ASK1 could facilitate its phosphorylation and activation, and a crosstalk may exist between phosphorylation and Lys29-linked polyubiquitination.

ASK1 is an important serine and threonine kinase involved in stress and immune response (*Shiizaki et al., 2013*). In *Map3k5$^{−/−}$* MEFs, TNFα-induced apoptosis and JNK activation are attenuated (*Tobiume et al., 2001*). In *Map3k5$^{−/−}$* thymocytes, FasL-induced activation of JNK and p38 is decreased while the apoptosis of thymocytes is not significantly affected (*Tobiume et al., 2001*). The primary neurons derived from *Map3k5$^{−/−}$* mice are resistant to polyglutamine-induced cell death (*Nishitoh et al., 2002*). More importantly, *Map3k5$^{−/−}$* mice demonstrate increased resistance to TLR4, but not TLR3 and TLR9, engagement, accompanied by impaired p38 activation (*Matsuzawa et al., 2005*). Upon infection with influenza virus, *Map3k5$^{−/−}$* MEFs show decreased JNK and p38 activation and inhibited cell death (*Maruoka et al., 2003*). Despite these convincing results, roles of ASK1 in innate antiviral response are not fully elucidated. Our study suggests that ASK1 activation by SCF$^{Fbxo21}$ is important in induction of type I IFN production, by using VSV and HSV-1 viruses as infection models. Our study also suggests that ASK1-mediated JNK activation, and to a lesser extent p38, is indispensable for type I IFN production, which is similar to the previous reports investigating TNFα-death receptor interaction (*Tobiume et al., 2001*). Activity of ASK1 is regulated by many molecules that complex with ASK1 (ASK1 signalosome, about 1,500–2,000 kDa), such as ASK1 itself, ASK1 homologues, thioredoxin, TRAF2, TRAF6 and Daxx etc. (*Shiizaki et al., 2013*). More unidentified molecules in ASK1 signalosome may exist for modulation of ASK1 activation. It has been proposed that ASK1 is usually activated via autophosphorylation after TRAF2− and TRAF6−assisted homodimerization in TNFα response (*Tobiume et al., 2002*; *Noguchi et al., 2005*).

In our study, we have found a complex containing Fbxo21 and SCF components, thus adding new components to ASK1 signalosome. Whether SCF$^{Fbxo21}$ associates with other components in ASK1 signalosome may need further investigation, and whether SCF$^{Fbxo21}$-mediated ASK1 activation needs TRAF2 and TRAF6 or upstream kinases may also be worthy of more studies.

When we are preparing the manuscript, it is reported that ASK1 is required for type I IFN production during antiviral innate response (*Okazaki et al., 2015*). In the current study, we also found that ASK1 deficiency led to impaired type I IFN production and increased virus replication. More importantly, we found that ASK1 deficient in Lys29-linkage could not rescue antiviral innate response in *Map3k5$^{-/-}$* RAW264.7 cells. Therefore, Fbxo21-mediated Ub-modification and activation of ASK1 are essential for innate antiviral response. It is interesting to find that JNK inhibitor could decrease the activation of IRF3 and type I IFN production. Previously it has been reported that JNK1/2 may be involved in activating IRF3 by phosphorylating IRF3 on Ser173 (*Zhang et al., 2009*) or licensing TBK1-mediated IRF3 phosphorylation on Ser396 (*Nociari et al., 2009*), which may explain the effects of JNK1/2 inhibition on decreased innate antiviral response mediated by Fbxo21. Therefore, it can be inferred that Fbxo21-mediated ASK1 activation may be involved in regulation of antiviral innate response via JNK-dependent IRF3 activation.

In sum, our study has shown that Fbxo21 plays an indispensable role in regulating innate antiviral response by promoting the Lys29-linkage and activation of ASK1 and subsequently ASK1-mediated JNK1/2 activation. Therefore Fbxo21-mediated regulation of ASK1 polyubiquitination and ASK1 activation may represent a new type of Ub modification by SCF complex and act as an intrinsic part of antiviral innate response.

## Materials and methods

### Mice, cells, reagents and antibodies

Wild type C57BL/6 mice (6–8 weeks old) were purchased from Joint Ventures Sipper BK Experimental Animal (Shanghai, China). All animal experiments were undertaken in accordance with the National Institute of Health Guide for the Care and Use of Laboratory Animals, and with the approval of the Scientific Investigation Board of Second Military Medical University, Shanghai. HEK293, RAW264.7 and L929 cells were obtained from ATCC (Manassas, VA) and cultured as recommended. Peritoneal macrophages and bone marrow-derived macrophages were obtained, prepared and cultured as described (*Tang et al., 2014*). Recombinant M-CSF, GM-CSF and IL-4 were from R&D Systems (Minneapolis, MN). Antibodies specific to Flag-tag (M2, ab49763), HA-tag (HA.C5, ab18181), His-tag (HIS.H, ab18184), Myc-tag (Myc.A7, ab18185), β-actin (AC-15, ab6276), c-fos (EPR883(2), ab134122), Cullin 1 (EPR3103Y, ab75817), Fbxo21 (EPR13163, ab179818), IKKβ (Y466, ab32135), and ubiquitin (Ubi-1, ab7254), and the agaroses used in immunoprecipitations were from Abcam Inc. (Cambridge, MA). Abs specific for ASK1 (D11C9, #8662), ERK1/2 (137F5, #4695), GSK3β (D5C5Z, #12456), IkBa (44D4, #4812), IRF3 (#4302), JNK1/2 (56G8, #9258), K48-linkage specific polyubiquitin (D9D5, #8081), K63-linkage specific polyubiquitin (D7A11, #5621), p38 (D13E1, #8690), p65 (L8F6, #6956), phospho-ASK1 (Thr845) (#3765), phospho-ERK1/2 (Thr202/Tyr204) (197G2, #4377), phospho-GSK3α/β (Ser21/9) (D17D2, #8566), phospho-IkBa (Ser32/Ser36) (5A5, #9246), phospho-IKKα/β (Ser176/Ser180) (C84E11, #2078), phospho-IRF3 (Ser396) (4D4G, #4947), phospho-JNK1/2 (Thr183/Tyr185) (81E11, #4668), phospho-p38 (Thr180/Tyr182) (12F8, #4631), Rbx1 (D3J5I, #11922) and Skp1 (D3J4N, #12248) were from Cell Signaling Technology (Beverly, MA). The inhibitors, including MLN-4924, SP600125 and SB203580, were obtained from MedChemExpress (Monmouth, NJ). C-class CpG ODN (ODN 2395), poly (dA:dT)/LyoVec and poly (dG:dC)/LyoVec were obtained from Invivogen (San Diego, California). Poly (I:C) and LPS (0111:B4) were purchased from Sigma (St. Louis, MO). The other non-specified reagents were form Sigma.

### Sequences, plasmids, transfection and RNA interference

The recombinant vectors encoding mouse ASK1 (GenBank No. NM_008580.4), Fbxo21 (GenBank No. NM_145564.3) and the indicated mutations were constructed as described (*Wang et al., 2009*; *Yang et al., 2011*). Flag-tagged vectors for Skp1, Cullin-1 and Rbx1 were from Origene (Beijing, China) and subcloned into pcDNA3.1 vector as described (*Wang et al., 2009*). For transient transfection of plasmids in RAW264.7 and L929 cells, the X-tremeGENE HP reagents were used according

to manufacturer's instructions (Roche, Welwyn Garden City, UK). For transient knockdown of *Map3k5* or *Fbxo21*, the siRNA duplexes specific for *Map3k5* (sc-29749; Santa Cruz Biotechnology, Dallas, TX) or *Fbxo21* (5' GCAGAAAGCTGGGTTAGAA 3') were transfected using the INTERFERin-HTS according to the standard protocol (Polyplus-transfection Company, Illkirch, France). The non-sense sequence 5'-TTCTCCGAACGTGTCACG-3' was used as control siRNA.

## CRISPR-Cas9-mediated depletion of Fbxo21 and ASK1

For the depletion of Fbxo21 or ASK1, pc3-U6-guide RNA-CMV-RED (encoding guiding RNA and red fluorescent protein) and Cas9-IRES-EGFP (encoding Cas9 and green fluorescent protein) plasmids (kind gifts from Shanghai Biomodel Organism Science & Technology Development Co., Shanghai, China) were cotransfected into RAW264.7 or L929 cells (*Tang et al., 2014*). Four target sequences for each guiding RNA synthesis against *Fbxo21* and *Map3k5* were tested, and the target sequences (5' GGTAGCGGGGGACAGCGCTA 3' for *Fbxo21*, and 5' TTTCGTTCAGCTTTTGAAGG 3' for *Map3k5*) were mostly efficient in depletion of Fbxo21 and ASK1. Cells with both red and green fluorescence were then sorted by using Gallios Flow Cytometer (Beckman Coulter, Brea, CA). Sorted cells were cultured for 3–5 days, and clones propagated from single cell were picked out. The depletion of Fbxo21 or ASK1 was confirmed by both Western blot and DNA sequencing.

## Quantitative-PCR

Total cellular RNA was extracted using Trizol reagent (Invitrogen Corporation, CA, USA), and cDNA was synthesized by using the AMV Reverse Transcriptase kit (Promega, Madison, WI). Specific primers used for Q-PCR assays and the mRNA quantification methods were as described (*Tang et al., 2014*). *β-actin* was used as the internal quantitative control.

## DNA microarray analysis

Microarray expression analysis was performed using a high-density oligonucleotide array (Affymetrix GeneChip array, Affymetrix, Santa Clara, CA, USA) as described (*Yang et al., 2015*), and microarray expression analysis was done according to the instruction manual. The extraction of total mRNA was done using Trizol (Invitrogen Corporation). The obtained mRNA was hybridized to each array. Hybridization of biotin-labeled cRNA fragment to Mouse Genome 430 2.0 array, washing, staining with streptavidin-phycoerythrin (Molecular Probes), and signal-amplification were performed according to the manufacturer's instructions. The changes of the interested genes were presented in heat map generated from the microarray results by using R Statistics.

## ELISA of cytokines

ELISA kits for mouse IFNβ and IL-6 were from R&D Systems (Minneapolis, MN). The concentrations of cytokines in the culture supernatants were determined as described (*Wang et al., 2009*; *Tang et al., 2014*).

## Luciferase assays

The AP-1, NF-κB, IRF3 and IFNβ luciferase reporter plasmids were obtained from Panomics of Affymetrix (Santa Clara, CA) or as described (*Wang et al., 2009*; *Yang et al., 2011*). Luciferase activities were measured with Dual-Luciferase Reporter Assay System (Promega, Madison, WI). The determination of reporter transactivation was performed as described previously (*Wang et al., 2009*; *Yang et al., 2011*).

## Determination of virus titer

For propagation of VSV and HSV-1, Vero cells were infected with virus at a MOI of 2, and the viruses produced by one replication cycle were harvested. For purification of virus, the cell cultures were first freeze-thawed twice, and the supernatants were clarified by centrifugation at 3000 ×g for 1 hr, followed by pelleting of the virus by ultracentrifugation at 45,000 ×g for 1 hr. For quantification of virus in infected cells, cells were homogenized in MEM supplemented with 2% FCS just before use. The homogenates were pelleted by centrifugation at 1600 ×g for 30 min, and the supernatants (100 μl) were used for plaque assays on monolayers of Vero cells. The cells were left overnight to settle and were infected by incubation for 1 hr at 37°C with serial dilutions. After 1 hr, 1 ml of

medium was added, and the plates were incubated for 2 days. After incubation, the plates were stained with 0.03% methylene blue to allow quantification of plaques. The numbers of plaque formation unit (PFU) were counted, and the results were expressed as PFU/ml of homogenates.

## Nanospray liquid chromatography–tandem mass spectrometry

Immunoprecipitated Fbxo21 immune complexes were separated on SDS-PAGE gel as described previously (*Yang et al., 2015*). After silver staining, each differential gel-band was excised and then analyzed by nano-ultra-performance liquid chromatography–electrospray ionization tandem mass spectrometry.

## Immunoprecipitation and immunoblot

The immunoprecipitations using anti-Fbxo21, anti-Flag or anti-Myc, and the immunoblot assays were performed as described previously (*Wang et al., 2009*; *Yang et al., 2015*).

## Nuclear protein extraction and DNA-bound transcription factor assay

Nuclear proteins were extracted by NE-PER Protein Extraction Reagent (Pierce, Rockford, IL) according to the manufacturer's instructions. Protein concentration was determined by the BCA protein assay (Pierce). For the analysis of nuclear transcription factors that could bind with specific DNA sequences, nuclear extracts were examined for c-fos, IRF3 and p65 by using the Trans[AM] Transcription Factor ELISA Kits from Active Motif Inc. (Carlsbad, CA) as recommended.

## Expression and purification of GST fusion proteins

Mouse wild-type or truncation coding sequences for *Fbxo21* were inserted 3' and in frame to glutathione S-transferase (GST) coding sequence in pGEX-2T vector (GE Health, Piscataway, NJ). *Escherichia coli* BL21 (DE3) cells (50 ml at an optical density 600 nm of 0.6) harboring the recombinant expression vectors were incubated with 1 mM isopropyl β-D-thiogalactopyranoside (IPTG; Sigma) at 37°C for 3 hr. For GST fusion protein purification, cells were harvested by centrifugation and suspended in 10 ml PBST containing 2 mM EDTA, 0.1% β-mercaptoethanol, 0.2 mM PMSF, and 5 mM benzamidine. Cells were lysed by passing through a French press twice at 1200 lb/in$^2$. 1 ml of bacterial supernatant was mixed with 100 μl 50% (v/v) glutathione-agarose beads (Pierce of Fisher, Rockford, IL) and incubated for 30 min at 4°C with gentle rotating. The agarose beads were washed four times with 10 ml ice-cold PBST or lysis buffer. The fusion proteins were eluted by 100 μl of 10 mM glutathione in 50 mM Tris (pH 8.0) at 4°C. The purity and quantity of fusion proteins were examined by SDS-PAGE followed by Coomassie blue staining.

## GST pull-down assay

The GST pull-down assays were conducted as described previously (*Wang et al., 2009*; *Yang et al., 2015*).

## In vitro kinase assay

For in vitro kinase assays, 100 μg proteins contained in total cell extracts or nuclear extracts were immunoprecipitated with indicated antibodies plus protein A/G beads by gently rocking at 4°C for 2 hr followed by centrifugation at 4°C for 5 min. The pellets were washed with the lysis buffer (20 mM Tris–HCl, pH 7.4, 150 mM NaCl, 1 mM EDTA, 1% Triton X-100, 2.5 mM sodium pyrophosphate, 1 mM β-glycerolphosphate, 1 mM Na$_3$VO4 and 1 mM PMSF). Each washed pellet was resuspended in the kinase assay buffer (25 mM Tris-HCl, pH 7.5, 5 mM β-glycerolphosphate, 2 mM DTT, 0.1 mM Na$_3$VO4, and 10 mM MgCl$_2$) and supplemented with 10 μM ATP and 0.5 μg of recombinant MKK4 (Abcam) in a total volume of 30 μl at 30°C for 30 min. Then phosphorylated MKK4 was examined by Western blot.

## Polyubiquitination assay

For analysis of ubiquitination of ASK1 in vivo, whole-cell extracts prepared with radioimmunoprecipitation assay buffer (50 mM Tris, pH 8.0, 150 mM NaCl, 1% (vol/vol) Nonidet-P40, 0.5% (wt/vol) sodium deoxycholate, 1% (wt/vol) SDS and proteinase inhibitors) supplemented with 10 mM N-

ethylmaleimide, were immunoprecipitated with indicated antibodies and analyzed by immunoblot as described (*Wang et al., 2009*; *Yang et al., 2015*).

The in vitro analysis of ubiquitination of ASK1 was performed as described (*Chen et al., 2013*) with modifications. The ubiquitination of ASK1 was performed in 50 μl buffer containing 40 mM Tris-HCl, pH 7.1, 40 mM NaCl, 1 mM β-ME, 5 mM $MgCl_2$ supplemented with 2 mM ATP, 1 μM ubiquitin aldehyde, 1 μg Conjugation Fraction A (F-340, BostonBiochem, Cambridge, MA), 1 μg/μl ubiquitin and derivatives (BostonBiochem), 10 ng/μl recombinant ASK1 (MAP3K5-9515M) and Cul1-Rbx1-Skp1 (CUL1-147H) (Creative Biomart, Shirley, NY) and 10 ng/μl GST-Fbxo21 in the presence or absence of Nedd8 Conjugation System (#J3150, UBPBio, Aurora, CO) and 5 μM MLN-4924 for 30 min at 37°C. Reaction products were finally processed for Ub immunoblotting.

## Statistical analysis

All the experiments were independently repeated at least three times. Dr. X.Z. who is blinded to group design performed the analysis. Results are given as mean ± SE or mean ± SD. Multiple comparisons were done with one-way ANOVA followed by Bonferonni's multiple comparison test. Statistical significance was determined as $p < 0.05$.

## GEO accession number

Complete microarray data set, GSE72077.

## Acknowledgements

We thank Mei Jin and Hao Shen for their excellent technical assistance. We thank Dr. Martin S.J. (Trinity College, Dublin, Ireland) for providing pGL3.5XκB-luciferase reporter plasmid, Dr. Wei Pan (Second Military Medical University) for VSV, Dr. Genhong Cheng (UCLA) for VSV-GFP, and Dr. Qihan Li for providing HSV-1 (F strain). We declare that we have no potential conflict of interests.

## Additional information

### Competing interests

XC: Reviewing editor, *eLife*. The other authors declare that no competing interests exist.

### Funding

| Funder | Grant reference number | Author |
| --- | --- | --- |
| National Key Basic Research Program of China | 2010CB911903 | Taoyong Chen |
| National Key Basic Research Program of China | 2013CB530500 | Xuetao Cao |
| National Natural Science Foundation of China | 81222039 | Taoyong Chen |
| National Natural Science Foundation of China | 81471566 | Taoyong Chen |
| National Natural Science Foundation of China | 81172851 | Taoyong Chen |
| National Natural Science Foundation of China | 31470874 | Jianli Wang |

The funders had no role in study design, data collection and interpretation, or the decision to submit the work for publication.

### Author contributions

ZY, Performed researches of Fbxo21 in innate immunity and performed the microarray assay and bioinformatics assay, Acquisition of data, Analysis and interpretation of data; TC, Designed and supervised research, performed researches of Fbxo21 in innate immunity, performed the microarray assay and bioinformatics assay, analyzed data and wrote the paper, Acquisition of data; XL,

Performed researches of Fbxo21 in innate immunity, Acquisition of data, Analysis and interpretation of data; MY, Performed researches of polyubiquitination assays, Acquisition of data; ST, Performed researches of Q-PCR assays, Acquisition of data; XZh, Performed the statistical analysis and cell cultures, Acquisition of data, Analysis and interpretation of data; YG, Assisted in FACS selection of Fbxo21-/- and Map3k5-/- cell lines, Acquisition of data; XS, Performed the microarray assay and bioinformatics assay, Acquisition of data; MX, Assisted in FACS selection of Fbxo21-/- and Map3k5-/- cell lines, Acquisition of data, Contributed unpublished essential data or reagents; WL, XZha, Assisted in MS assays and interpretation of MS data, Acquisition of data, Contributed unpublished essential data or reagents; QW, Performed the ELISA assays, Acquisition of data; XC, Designed and supervised research, analyzed data and wrote the paper; JW, Designed and supervised research, analyzed data, wrote the paper, Acquisition of data

### Author ORCIDs
Taoyong Chen, http://orcid.org/0000-0002-2507-0027

### Ethics

Animal experimentation: This study was performed in strict accordance with the recommendations in the Guide for the Care and Use of Laboratory Animals of the National Institutes of Health, and was approved by the Scientific Investigation Board of Second Military Medical University, Shanghai (Case No. SMMU-2015-0067).

## Additional files

### Major datasets

The following dataset was generated:

| Author(s) | Year | Dataset title | Dataset URL | Database, license, and accessibility information |
|-----------|------|---------------|-------------|---------------------------------------------------|
| Chen T, Yu Z, Cao X | 2015 | Gene expression in RAW264.7 cells infected with VSV and HSV-1 | http://www.ncbi.nlm.nih.gov/geo/query/acc.cgi?acc=GSE72077 | Publicly available at the NCBI Gene Expression Omnibus (accession no: GSE72077). |

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
