## [Decision Letter]

Thank you for submitting your work entitled "Lys29-linkage of ASK1 by SCF^Fbxo21^ is required for antiviral innate response" for consideration by *eLife*. Your article has been reviewed by two peer reviewers, Hidenori Ichijo and Michaela Gack, and the evaluation has been overseen by Ruslan Medzhitov as the Reviewing Editor and Tadatsugu Taniguchi as the Senior Editor.

The reviewers have discussed the reviews with one another and the Reviewing Editor has drafted this decision to help you prepare a revised submission.

Summary:

In this manuscript, Yu et al. show that the F-box only protein, Fbxo21, is a component of the E3 ubiquitin ligase SCF complex and is critical for the innate immune response to viral infection. A microarray-based screen of Fbxo protein expression levels during viral infection identified Fbxo21 as being potentially involved in innate immune signaling. Using Fbxo21 knockout cells, the authors show that Fbxo21 plays an important role in the activation of innate immune signaling, specifically the production of cytokines and type I interferon, in response to viral infection. Using mass spectrometry analysis, the authors identify the protein kinase ASK1 as an interaction partner of Fbxo21. Through extensive characterization of ASK1 ubiquitination by SCF^Fbxo21^, the authors conclude that Fbxo21 acts by targeting six lysine residues (Lys946, Lys950, Lys951, Lys952, Lys953 and Lys957) in ASK1 for SCF-mediated non-degradative K29-linked ubiquitination. Finally, the authors provide data which show that K29-linked ubiquitination of ASK1 is critical for downstream activation of JNK1/2, p38, and IRF3 and thus cytokine responses to infection with VSV and HSV-1. The authors propose a model in which Fbxo21 targets ASK1 for K29-linked ubiquitination by the SCF complex, which enables ASK1 to activate various innate immune signaling molecules, ultimately resulting in the transcription of proinflammatory cytokines and interferons.

Essential revisions:

1) Although it is convincing that Fbxo21 has an ability to lead Lys29-linked ubiquitination of ASK1 shown in Figure 6, it is still not clear if Fbxo21-dependent Lys29-linked ubiquitination is required for ASK1 activity. Although ASK1 K13 mutant exhibited lower kinase activity, mutated sites in ASK1 K13 mutant might form different types of ubiquitin chain such as K48-linked or K63-linked, which might be involved in the regulation of ASK1 activity. To overcome this issue, the authors should measure ASK1 activity in Fbxo21-knockout cells and in MLN-4924-treated cells with or without virus infection. Moreover, the time course of ubiquitination and activation of ASK1 should be examined to discuss the requirement of ubiquitination for activation.

2) I recommend one additional experiment to strengthen the authors' model that Fbxo21 is a component of the SCF complex during viral infection.

The experiments intended to demonstrate that Fbxo21 is part of the SCF complex were performed by overexpressing tagged forms of individual SCF components. Performing an endogenous IP of SCF components and Fbxo21 in uninfected and virus-infected cells would strengthen the authors' point that Fbxo21 is indeed part of the SCF complex during viral infection.

---

## [Author Response]

Essential revisions:

*1) Although it is convincing that Fbxo21 has an ability to lead Lys29-linked ubiquitination of ASK1 shown in Figure 6, it is still not clear if Fbxo21-dependent Lys29-linked ubiquitination is required for ASK1 activity. Although ASK1 K13 mutant exhibited lower kinase activity, mutated sites in ASK1 K13 mutant might form different types of ubiquitin chain such as K48-linked or K63-linked, which might be involved in the regulation of ASK1 activity. To overcome this issue, the authors should measure ASK1 activity in Fbxo21-knockout cells and in MLN-4924-treated cells with or without virus infection. Moreover, the time course of ubiquitination and activation of ASK1 should be examined to discuss the requirement of ubiquitination for activation.*

The net effects of Fbxo21 deficiency on Lys29-linkage of ASK1 have not been directly demonstrated because of the lack of Lys29-specific antibody. To address this concern, we examined the phosphorylation and kinase activity of ASK1 in *Fbxo21*^+/+^ or *Fbxo21*^–/–^ RAW264.7 cells upon virus infection in the presence or absence of MLN-4924 as suggested. Moreover, we also examined the dynamic poly-Ub modification (using total Ub or K48- and K63-specific antibodies) and dynamic activation of ASK1 in *Fbxo21*^+/+^ or *Fbxo21*^–/–^ RAW264.7 cells. These new data are now presented in Figure 7—figure supplement 3 and described inthe last two paragraphs of the subsection “Lys29-linkage of ASK1 by SCF^Fbxo21^ is required for ASK1 activation”.

2) I recommend one additional experiment to strengthen the authors' model that Fbxo21 is a component of the SCF complex during viral infection.

The experiments intended to demonstrate that Fbxo21 is part of the SCF complex were performed by overexpressing tagged forms of individual SCF components. Performing an endogenous IP of SCF components and Fbxo21 in uninfected and virus-infected cells would strengthen the authors' point that Fbxo21 is indeed part of the SCF complex during viral infection.

As suggested, we performed the assays of endogenous interactions of Fbxo21 with Skp1, Cul1 and Rbx1 in wild type RAW264.7 cells before and after VSV or HSV-1 infection. The data are now presented in Figure 5—figure supplement 2and described in the last paragraph of the subsection “Fbxo21 is a component of the SCF complex and interacts with ASK1“.